# Human Primary Astrocytes Differently Respond to Pro- and Anti-Inflammatory Stimuli

**DOI:** 10.3390/biomedicines10081769

**Published:** 2022-07-22

**Authors:** Piotr Szpakowski, Dominika Ksiazek-Winiarek, Malgorzata Turniak-Kusy, Ilona Pacan, Andrzej Glabinski

**Affiliations:** Department of Neurology and Stroke, Medical University of Lodz, ul. Zeromskiego 113, 90-549 Lodz, Poland; dominika.ksiazek@umed.lodz.pl (D.K.-W.); m.turniak.kusy@gmail.com (M.T.-K.); ilona.pacan@stud.umed.lodz.pl (I.P.); andrzej.glabinski@umed.lodz.pl (A.G.)

**Keywords:** astrocytes, phenotypes, chemokines, cytokines, neuroinflammation

## Abstract

For a long time, astrocytes were considered a passive brain cell population. However, recently, many studies have shown that their role in the central nervous system (CNS) is more active. Previously, it was stated that there are two main functional phenotypes of astrocytes. However, nowadays, it is clear that there is rather a broad spectrum of these phenotypes. The major goal of this study was to evaluate the production of some inflammatory chemokines and neurotrophic factors by primary human astrocytes after pro- or anti-inflammatory stimulation. We observed that only astrocytes induced by inflammatory mediators TNFα/IL-1a/C1q produced CXCL10, CCL1, and CXCL13 chemokines. Unstimulated astrocytes and those cultured with anti-inflammatory cytokines (IL-4, IL-10, or TGF-β1) did not produce these chemokines. Interestingly, astrocytes cultured in proinflammatory conditions significantly decreased the release of neurotrophic factor PDGF-A, as compared to unstimulated astrocytes. However, in response to anti-inflammatory cytokine TGF-β1, astrocytes significantly increased PDGF-A production compared to the medium alone. The production of another studied neurotrophic factor BDNF was not influenced by pro- or anti-inflammatory stimulation. The secretory response was accompanied by changes in HLA-DR, CD83, and GFAP expression. Our study confirms that astrocytes differentially respond to pro- and anti-inflammatory stimuli, especially to inflammatory cytokines TNF-α, IL-1a, and C1q, suggesting their role in leukocyte recruitment.

## 1. Introduction

Astrocytes are the major glial cell population of the central nervous system (CNS), defined by their stellate morphology and expression of the glial fibrillary acidic protein (GFAP) [1]. These cells possess a high rate of metabolic activity and a wide range of functions that are crucial to maintaining a balanced brain microenvironment. Astrocytes are classically divided into three major types based on their morphology and spatial organization: protoplastic astrocytes in gray matter, fibrous astrocytes in white matter, and radial astrocytes surrounding ventricles.

Astrocytes, long considered mainly as a trophic and mechanical support for neurons, have progressively gained more attention as their role in CNS pathologies has become more obvious. They actively regulate synaptic transmission via the release and clearance of neurotransmitters and the regulation of the extracellular ion concentration [2,3]. They contribute to the formation and maintenance of the blood–brain barrier (BBB) [4], which separates the peripheral blood circulation from the CNS [5]. Moreover, astrocytes secrete various neurotrophic factors to regulate synaptogenesis, neuronal differentiation, and survival [6,7]. It is suggested that they may also act as antigen-presenting cells, regulating immune responses within the CNS [8]. Additionally, they may mediate the transmigration of reactive T cells into the CNS via the cell-adhesion molecules, chemokines, and matrix metalloproteinases [9]. The phenotypic status of astrocytes is a widely discussed issue because its understanding is critical for the study of the mechanisms involved in CNS lesions and repair. For a long time, there was a debate regarding whether there are two separate functional phenotypes of astrocytes, or whether there is a continuum of various mixed subtypes of these cells ranging from the resting to reactive state [10,11,12]. Nowadays, researchers agree with the second hypothesis, which is supported by the results of numerous studies [13]. Astrocytes may respond to and secrete numerous factors, e.g., cytokines, chemokines, trophic factors, and others [14]. Their response to injury may be directed at the protection/regeneration of the CNS, and also at the escalation of detrimental processes. The reparative or destructive outcome depends on the molecular context of the CNS microenvironment. The mechanisms translating injury-derived signals into an astroglial reactive phenotype are complex, and yet largely unknown. It is now clear that astrocytes represent a complex, heterogeneous, and functionally diverse population of cells [15]. Not all astrocytes express the same proteins or receptors, and so they may respond to the same factors in a distinct manner [15].

It is increasingly clear that the changes and dynamics of the astrocytes’ response to damage are crucial for the various CNS pathologies’ outcomes. Here, we seek to assess the functional phenotype of human astrocytes in response to several stimulatory factors, and the secretory activity of these cells in various in vitro conditions.

## 2. Materials and Methods

### 2.1. Primary Astrocyte Cell Cultures

Cerebral cortex astrocytes isolated from the human brain were obtained from four 18–22-week-old donors and purchased from ScienCell Research Laboratories Inc. (San Diego, CA, USA). Cell cultures were grown on poly-L-lysine-coated 75 cm^2^ culture flasks in commercially available astrocyte culture medium supplemented with penicillin, streptomycin, 2% fetal bovine serum (FBS), and astrocyte growth supplement (all components from ScienCell Research Laboratories) in 5% CO_2_ and at 37 °C, in an increased-humidity atmosphere. The culture medium was changed every 3 days (confluency < 70%) or daily (for confluency 70–90%). At 90% confluency, cells were harvested with 0.025% Trypsin/EDTA (ethylenediaminetetraacetic acid) in Dulbecco’s PBS, washed with PBS/10% FBS, centrifuged (150× *g*, 5 min, 20 °C) and seeded on poly-L-lysine-covered 48- or 6-well culture plates (respectively, 2.5 × 10^4^ cells/well or 1.3 × 10^6^/well), and cultured for 24 h (to allow attachment to the growth surface) before stimulation with cytokines. 

### 2.2. Analysis of Astrocytes’ Response to Cytokines

Astrocytes were stimulated in culture medium with a cocktail of proinflammatory cytokines: rhTNF-α (30 ng/mL, R&D Systems, Ixonia, WI, USA), rhIL-1a (3 ng/mL, R&D Systems), hC1q (400 ng/mL, MyBiosource, San Diego, CA, USA). These cytokines are released by microglia cells and were reported to have potential in induction of proinflammatory and neurotoxic astrocytes’ phenotype [16]. For alternative stimulation, cells were cultured in medium containing rhIL-10 (10 ng/mL, R&D Systems), IL-4 (1 ng/mL R&D Systems), or TGF-β (10 ng/mL, R&D Systems) as these cytokines possess anti-inflammatory potential and were reported to be present in the brain during inflammation [17,18]. Non-stimulated astrocytes cultured in medium alone were also included as a control. After 6-day stimulation on 48-well plates, cell cultures were examined using a microscope, and the culture medium was collected, centrifuged (5000× *g*, 10 min, 20 °C), aliquoted, and stored at −80 °C for further measurements of chemokine and neurotrophin production. Concentrations of CCL1, CXCL1, CXCL13 chemokines, and neurothrophins—platelet-derived growth factor (PDGF-AA), brain-derived neurotrophic factor (BDNF), glial-derived neurotrophic factor (GDNF), and beta-nerve growth factor (β-NGF) were determined with ELISA assay (DuoSet, R&D Systems); production of IL-1β and CXCL10 was assessed with ELISA kits from Biolegend. All measurements were performed according to optimized protocols provided with the ELISA sets by the manufacturers.

### 2.3. Analysis of Surface Receptor Expression and Intracellular GFAP Level with Flow Cytometry

For flow cytometry analysis, astrocytes were harvested from 6-well plates after 48 h stimulation with investigated cytokines. Cells were collected on ice with cell dissociation solution (Sigma, St. Louis, MO, USA) and washed with cold PBS/1%FBS. For multicolor staining of surface receptors, a panel of the following fluorochrome-labeled antibodies was applied: anti-CD80 APC mouse IgG1 κ, anti-CD83 PE mouse IgG1 κ, anti-CD86 Alexa Fluor488 mouse IgG2b κ, anti-HLA-DR Alexa Fluor 700 mouse IgG2a κ (all antibodies from Biolegend, San Diego, CA, USA). After incubation with fluorescent antibodies (30 min, 4 °C, dark), cells were washed two times with cold PBS/1%FBS, fixed with formalin (20 min, 4 °C), washed two times with cold PBS/1%FBS, and frozen in FBS containing 10%DMSO (dimethyl sulfoxide) for further analysis. Flow cytometry measurements were conducted on an LSR II flow cytometer (Becton Dickinson, Franklin Lakes, NJ, USA). For GFAP staining, cells were fixed with formalin (20 min, 4 °C), permeabilized with Perm/Wash solution (BD Pharmingen), and stained with anti-GFAP PE antibody (BD Pharmingen). For all staining procedures, samples with antibody isotype controls were also utilized.

### 2.4. Statistical Analysis

Statistical analysis was carried out using Statistica 13.1 software (TIBCO Software Inc., Houston, TX, USA). The normality of distribution was checked with the Shapiro–Wilk test. Variables with normal distribution were analyzed with a parametric one-way ANOVA test followed by post hoc Tukey’s honest significant difference test. Variables with abnormal distribution were analyzed with a non-parametric Mann–Whitney U test. Results from flow cytometry measurements were analyzed with the Wilcoxon signed-rank test. Statistical differences were considered significant for *p* values < 0.05.

## 3. Results

### 3.1. Proinflammatory Environment Results in Elevated GFAP Level in Astrocytes

To confirm that the analyzed cells were astrocytes, we measured the intracellular level of the astrocyte marker—the GFAP protein—by flow cytometry. All analyzed cells showed a high intracellular GFAP protein level (Figure 1A). GFAP increased in astrocytes exposed to the microglia proinflammatory cytokine mixture (TNF-α/IL-1a/C1q) compared to unstimulated cells (*p* = 0.02) (Figure 1B) [16]. We did not notice any significant changes in GFAP level in cells stimulated with other cytokines. Cells cultured in medium alone were used as a control. 

### 3.2. CD83 and HLA-DR Molecules Are Upregulated in Astrocytes Exposed to TNF-α/IL-1a/C1q Cytokines

Analysis of surface receptors’ expression revealed weak density and minimal changes in CD80 and CD86 molecules on cells exposed to the analyzed cytokines. Interestingly, we noticed a high level of CD83 on the surfaces of the analyzed cells, and the expression, measured as the median fluorescence intensity, significantly increased (*p* = 0.03) on cells exposed to TNF-α/IL-1a/C1q compared to cells cultured in medium alone (Figure 2, Appendix A).

### 3.3. Proinflammatory Stimuli Induced Dramatic Chemokine Release in Astrocytic Cultures

We observed strong CCL1 (*p* = 0.00001), CXCL1 (*p* = 0.0000001), CXCL10 (*p* = 0.000001) and CXCL13 (*p* = 0.00001) chemokine production in astrocyte cultures, as well as the induction of proinflammatory IL-1β (*p* = 0.02), in response to the TNF-α/IL-1a/C1q cytokine cocktail in cell culture supernatants. These chemokines were not detected in supernatants from unstimulated cells or astrocytes cultured with anti-inflammatory cytokines IL-4, IL-10, or TGF-β1 (Figure 3). 

### 3.4. Various Cytokine Environments Differently Regulate PDGF-A Expression in Astrocytes

Unstimulated astrocytes cultured for 6 days in an astrocyte growth medium spontaneously produced high amounts of PDGF-A (mean 3109 ± 500 pg/mL). The addition of IL-4 or IL-10 to the culture medium did not affect the PDGF-A level in collected supernatants. Cells cultured in proinflammatory conditions showed significantly decreased PDGF-A release, as compared to unstimulated astrocytes (*p* = 0.0004). However, in response to TGF-β1, astrocytes showed significantly increased PDGF-A production compared to medium alone (*p* < 0.003) and compared to proinflammatory conditions (*p* = 0.0017) (Figure 4A).

BDNF production in astrocyte cultures was also detected; however, we did not observe the impact of stimulatory conditions on BDNF levels (Figure 4B). GDNF production was detected only in the cell culture of one donor, and β-NGF was not detectable in 6-day cultures with the applied ELISA kits (data not shown).

## 4. Discussion

Several studies have indicated the existence of the neuroimmune system in the CNS and its role in CNS functioning, homeostasis, and pathology [19,20]. The main cellular components of this system are glial cells—astrocytes and microglia—which initiate communication with other cell types via the production of various signaling molecules. Astrocytes are the most numerous glial cell type in the CNS [21]. They play various roles in the CNS, as regulators of the physiological state and responders to various pathological conditions, such as injury or infection [22]. The CNS is considered an immune-privileged region due to the presence of the BBB, which limits access to peripheral antibodies and leukocytes [23]. Astrocytes, as part of the BBB, are amongst the first CNS-resident cells that come into contact with blood-derived leukocytes entering the brain during neuroinflammation [4]. Moreover, the anatomical localization of astrocytic endfeet enables them to react to various soluble factors in the meningeal space [24,25]. The reciprocal interaction between activated peripheral immune cells and astrocytes impacts the active migration of leukocytes into the CNS [26].

Astrocytes secrete two main chemokines controlling the recruitment of perivascular leukocytes into the CNS [27]. One of these is CXCL10, which is known as a potent chemoattractant for Th1 cells, NK cells, and monocytes/macrophages [28,29]. CXCL10 is induced locally in the CNS in diverse pathologic states, e.g., Alzheimer’s disease [30] and multiple sclerosis (MS) [31]. An increase in mRNA encoding CXCL10 in experimental autoimmune encephalomyelitis (EAE)-affected mouse brains under an inflammatory state was related to an increase in GFAP expression and astrogliosis [32]. Studies by Oh and others established that CXCL10 gene expression by astrocytes is quite dynamic and can be regulated by a variety of factors, e.g., IL-1β, TNF, and LPS [33,34]. Our results add to this group additional factors such as IL-1a and C1q (Figure 3C), similarly to what was observed by Liddelow et al. [16]. In our studies, only astrocytes stimulated with proinflammatory cytokine cocktail TNF-α/IL-1a/C1q were able to secrete CXCL10. There was no detectable CXCL10 production in unstimulated astrocytes or in those stimulated to induce an alternative protective phenotype (Figure 3C). This result is in agreement with previous reports on CXCL10 expression in the rodent CNS. Significant elevation of this chemokine has been observed in diverse neuropathologies, including inflammatory diseases such as EAE, contusion injury, cerebral ischemia, and neurotoxicant-induced neurodegeneration [35,36,37,38,39].

It has been observed that astrocyte-produced CXCL10 regulates not only leukocytes’ accumulation but also microglial migration toward an injury site, through a CXCR3-mediated mechanism [27,35,40,41,42,43]. This CXCL10-induced microglial movement has been linked to efficient myelin debris clearance in a cuprizone-induced demyelination model [44]. Moreover, in an in vitro model of myelination, astrocytes with high CXCL10 expression were unable to promote this process [45]. The addition of this chemokine to normal myelinating cultures leads to reduced myelination of axons. This observation overall points to CXCL10’s function in oligodendrocyte maturation and axonal wrapping [45]. In our study, astrocytes with the proinflammatory phenotype secreted this chemokine on a high level, suggesting the possibly important role of these cells in the above-described processes (Figure 3C).

CCL1 is a chemokine that induces chemotaxis and plays an important role in the regulation of apoptosis [46]. This chemokine exerts its effects via the CCR8 receptor [47], whose constitutive expression has been shown in monocytes and macrophages, Th2 and Treg lymphocytes [48,49,50], NK cells, and immature B cells. It has been reported that the CCL1/CCR8 pathway is associated with phagocytic macrophages and activated microglia in active lesions in MS, and the level of CCL1 directly correlates with demyelinating activity. High expression of CCL1 and CCR8 in the CNS during EAE suggests that CCL1 plays an important role in the neuroinflammation process [50,51].

In addition, in in vivo studies, CCL1 increased the number of GFAP-positive astrocytes and Iba-1-positive microglia [52]. Moreover, it has been shown, in an in vivo model of ischemia, that CCL1 produced by astrocytes and oligodendrocytes attracts Treg cells to the ischemic brain [53]. Reactive microglia exert direct neurotoxic effects, but they also participate in neuronal injury indirectly. Increased secretion of cytokines and chemokines, e.g., CCL1, CCL20, IL-1β, IL-6, and TNF-α, is a hallmark of the proinflammatory activation phenotype of microglia. Some of these chemokines and cytokines (e.g., IL-1α, TNF-α and complement C1q) are able to induce the proinflammatory phenotype of astrocytes [54]. We observed that the stimulation of astrocytes by the mixture of IL-1α, TNF-α and complement C1q stimulated CCL1 production by these cells (Figure 3A).

CXCL13 is constitutively expressed in lymphoid organs and has been shown to be a key chemokine in lymphocyte recruitment and compartmentalization. CXCL13 exerts its effect via the CXCR5 receptor [55]. The function of CXCL13 is only partially defined and mainly related to B cell chemoattraction to the CNS [56]. However, CXCL13’s effects are not limited to the development and support of lymphoid tissues—it is also involved in chronic inflammation through the formation of tertiary lymphoid structures (TLS) [57].

CXCL13 is not expressed in the CNS in physiological conditions, but its expression is high in the brain and spinal cord under pathological conditions, such as autoimmune demyelination, primary CNS lymphoma, and Lyme neuroborreliosis (LMN) [58,59,60]. It has been suggested that the sources of CXCL13 in the CNS are as follows: monocytes in LMN, macrophages infiltrating lesions, perivascular stromal cells in primary CNS lymphoma, microglial cells, or meningeal TLS in MS [56,61,62]. The fact that astrocytes are able to produce CXCL13 upon activation by proinflammatory cytokines has not been reported before.

It was observed in in vitro studies that CXCL13 was produced by monocytes and by macrophages (at a much higher level). Expression of CXCL13 (both mRNA and protein) was induced by TNF-α and IL-1β but inhibited by IL-4 and IFN-γ [56]. In our study, CXCL13 production by astrocytes was induced by incubation with a mixture of TNF-α, IL-1a, and C1q only (Figure 3D).

Various trophic factors released by astrocytes impact neuronal survival and plasticity after brain injury. These factors play important roles in pathological conditions, where they trophically support damaged neurons and oligodendrocytes and, some of them activate progenitor cells [11]. Moreover, growth factors also act on astrocytes in an autocrine/paracrine manner, thus contributing to a feed-forward amplification loop, which starts and sustains reactive astrogliosis [63,64].

One of the most studied trophic factors is the platelet-derived growth factor (PDGF). The PDGF family involves five proteins, existing as homodimers of chains from A to D, and as one known heterodimeric form (AB). PDGF molecules are ubiquitous in mammalian brains, where they are involved in the regulation of neuronal system development, although, in the adult brain, PDGF family members are implicated in numerous cellular activities. PDGF-A regulates oligodendrocyte precursor cell (OPCs) development, proliferation, and survival, determining the number of oligodendrocytes in the developing [65] and adult brain [66]. PDGF-A regulates also the proliferation and branching of astrocyte cells [67].

In our experimental conditions, astrocytes cultured in a medium without any stimulus spontaneously produced PDGF-A. Cytokines with anti-inflammatory activity—IL-10 and IL-4—were not able to alter PDGF-A production in astrocyte cultures compared to the quiescent state. TGF-β1 strongly enhanced PDGF-A production in astrocytic cultures, whereas cells cultured in proinflammatory conditions showed lower production of this neurotrophin (Figure 4A). Our observation is partially similar to the results described by Silberstein et al. [68]. The researchers reported mRNA for PDGF-A in untreated cells at a detectable level; however, its production in culture did not increase in response to TGF-β1 stimulation. Moreover, the authors also described an increased PDGF response to TNF-α, which is a strong proinflammatory agent. In their research, however, they used astrocyte-enriched, but not pure, astrocyte cultures.

The production of PDGF-A by astrocytes and its increase in response to TGF-β1 may be especially significant for functional relapses in inflammatory demyelinating disorders such as multiple sclerosis. PDGF-A-expressing astrocytes are able to stimulate myelin renewal due to the PDGF-A-dependent activation of oligodendrocytes and their precursors [69]. In experiments utilizing transgenic mice with expression of the human PDGF-A gene over the control of a specific promoter, remyelination in a cuprizone-induced MS model was associated with the increased density of oligodendrocyte progenitor cells and a reduced apoptosis ratio as compared to control animals [70].

During the MS course, demyelination is associated with inflammation development in the CNS. Strong proinflammatory conditions provided by activated microglia induce a neurotoxic astrocyte phenotype through TNF-α/IL-1a/C1q-dependent signaling [16], which, in turn, according to our results, may lead to a decrease in PDGF-A secretion (Figure 4A). This results in demyelination and less efficient remyelination, especially during a chronic inflammatory reaction, characteristic of MS.

Brain-derived neurotrophic factor (BDNF) plays an important role in neural survival, synaptic plasticity, and long-term potentiation (LTP); thus, its alterations are correlated with cognitive impairments [71,72]. It is also important for dendrite outgrowth and spine number, which has been observed in an in vitro study [73]. BDNF is an important factor for OPCs. It promotes their proliferation and differentiation into mature oligodendrocytes [74]. It may also impact the differentiation of neural stem/progenitor cells into oligodendrocyte lineage cells [75]. Astrocytes are able to express BDNF and release it, as well as recycle and store this neurotrophin for use in an activity-dependent manner [76,77]. Moreover, BDNF promotes astrocytes’ proliferation and survival through its truncated form of receptor tropomyosin receptor kinase B (TrkB) located on these cells, pointing to the existence of a feed-forward regulatory loop [78].

Astrocytes are known to express BDNF following injury in vivo [79]. It was shown that astrocyte-derived BDNF may be a source of trophic support, which can reverse deficits present following demyelination [80]. It was observed that, during endogenous recovery from ischemic injury of white matter, astrocytes support the maturation of OPCs by secreting BDNF [81]. Additionally, transgenic mice with downregulated expression of BDNF in GFAP-positive astrocytes subjected to ischemic injury exhibited a lower number of newly generated oligodendrocytes and larger white matter damage [81].

However, in our experimental conditions, there was no significant change in BDNF expression levels (Figure 4B). This may be due to the stimulatory factors used in our research.

Glial-derived neurotrophic factor (GDNF) is an important regulator of neurons’ growth and differentiation, and its expression is elevated during brain development [82,83]. In the healthy developed brain, neurons are the major source of GDNF; however, in inflammation, caused by infections or brain injury, astrocytes, as well as microglia cells, participate in GDNF production [84,85,86,87,88,89]. Elevated production of GDNF in the brain supports the renewal of injured tissues [90]. Astrocyte cells, through GDNF secretion, are able to abolish microglia activation by Zymosan A, which was shown on midbrain astrocyte cultures collected from the four-day-old pups of Wistar rats [91].

In our research, we did not observe the production of GDNF by astrocytes in either one- or six-day cultures (except one donor, data not shown). In our stimulation model, we used a limited number of cytokines; moreover, we did not use any danger signals, which may be necessary as tissue damage induces GDNF production. Brambilla et al. reported the role of TNF-α-TNFR1 signaling in the control of GDNF synthesis in spinal cord astrocytes in SOD1 mice [92]. In our study, TNF-α, together with IL-1a and C1q, was not able to induce GDNF release at a detectable level. This might be explained by the functional diversity of astrocyte subtypes, resulting in the absence of the proper form of the TNF-α receptor (TNFR2 in spite of TNFR1), as well as the participation of other cells and processes in providing signals for astrocytes after TNF-α injection.

Surprisingly, we did not observe nerve growth factor (NGF) production in astrocyte cultures in either one- or six-day stimulations at ELISA detection level (data not shown), although astrocytes are considered major producers of this neurotrophic factor [93]. Metoda et al. described the role of histamine together with IL-1β in the induction of NGF production in cortical astrocytes from rats [94]. Another inducer of NGF is IFN-β, which was able to induce a 40-fold increase in mRNA for NGF [95]. It can be stated that the lack of NGF production in our study may be the result of the detection assay or more likely, due to the stimulation conditions. Our results suggest that astrocytes do not produce NGF in response to the examined pro- and anti-inflammatory factors.

## 5. Conclusions

In our study, we observed the differences in the secretory activity of astrocytes after stimulation with selected factors. There were differences not only between the modes of action of pro- and anti-inflammatory molecules but also among these groups. This indicates the complexity of the regulation of astrocytes’ functional phenotypes. Moreover, to the best of our knowledge, this is the first study reporting the production of CXCL13 by astrocytes stimulated with proinflammatory factors. As astrocytes may contribute to both pathological and repair processes, this is enormously important to deepen the knowledge about the regulation of their functions, especially in CNS pathologies.

## Figures and Tables

**Figure 1 biomedicines-10-01769-f001:**
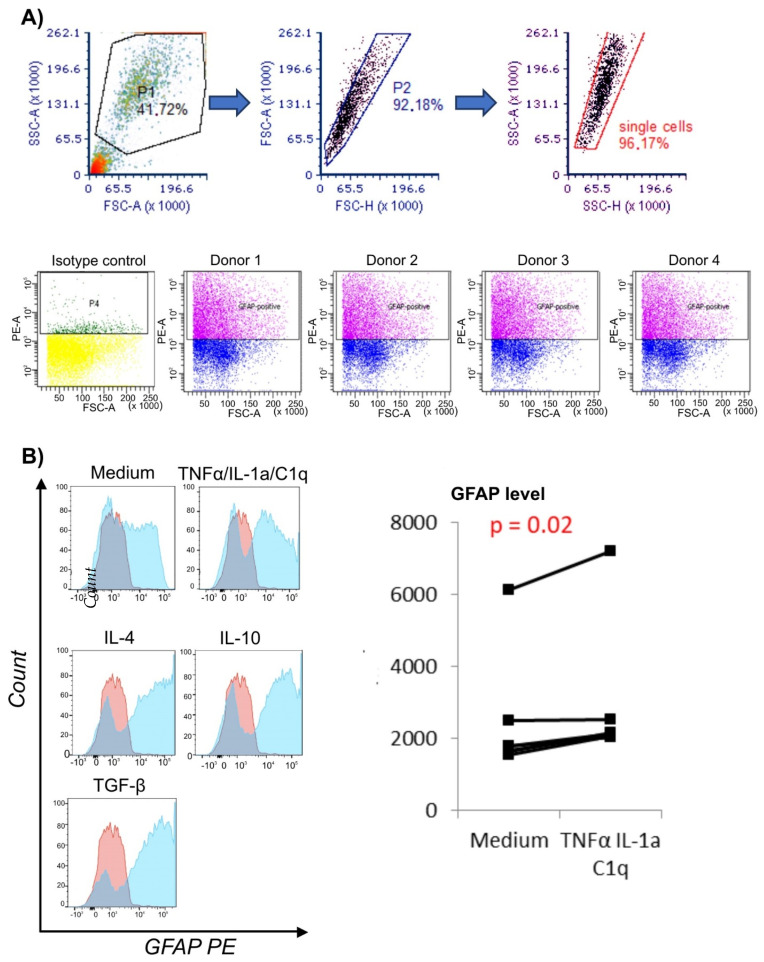
Flow cytometric analysis of human primary astrocyte cells. (**A**) Expression of glial fibrillary acidic protein (GFAP)—2D graphs of GFAP-positive astrocyte cells and gating strategy. (**B**) Cytokine-induced changes in GFAP expression—representative histograms from 1 donor. Red-filled histograms represent cells stained intracellularly with isotype control antibodies; blue histograms represent cells stained with GFAP-specific antibodies conjugated with PE. Combined results from 4 donors showing increased GFAP level are presented on the graph as changes in MFI (median fluorescence intensity). Statistically significant differences and *p* values were assessed with Wilcoxon signed-rank test, and *p* < 0.05 was considered significant.

**Figure 2 biomedicines-10-01769-f002:**
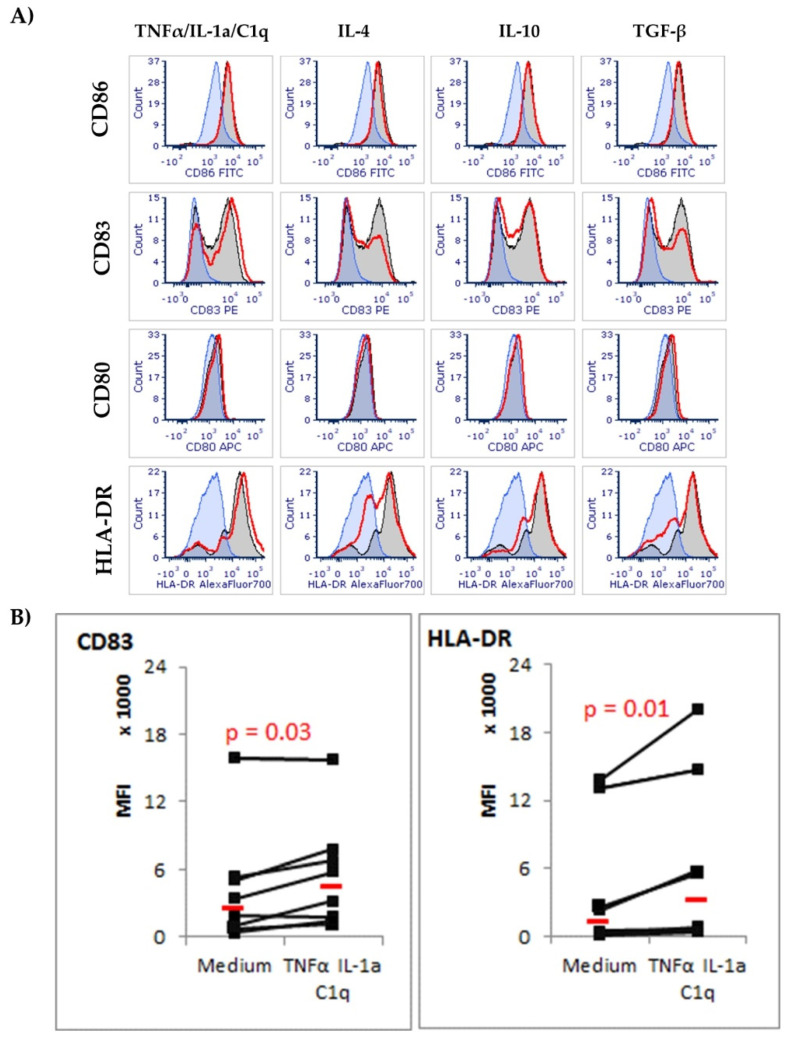
Expression of surface co-stimulatory CD86, CD83, CD80, and MHC class II (HLA-DR) molecules on astrocyte cells in response to cytokine environment. (**A**) Panel of histograms for one representative donor; blue-filled histograms represent cells stained with isotype control antibodies, grey-filled histograms represent molecule expression on astrocytes grown in non-stimulatory conditions, red line histograms represent molecule expression on astrocytes grown in cytokine-enriched conditions. (**B**) Combined results for 4 different donors in 2 independent experiments. Normality of distribution was assessed with Shapiro–Wilk test. Statistically significant differences and *p* values were assessed with Wilcoxon signed-rank test, and *p* < 0.05 was considered as significant.

**Figure 3 biomedicines-10-01769-f003:**
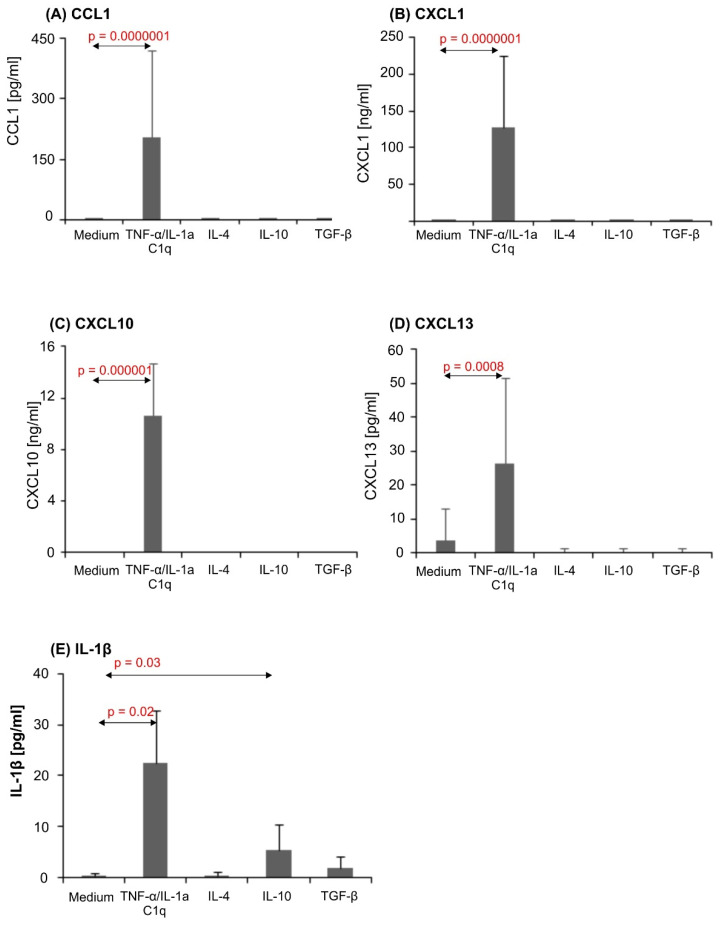
CCL1 (**A**), CXCL1 (**B**), CXCL10 (**C**), CXCL13 (**D**), and IL-1β (**E**) production in human astrocyte cultures. Results from at least 2 separate experiments conducted on primary astrocyte cells, collected from 4 human donors. Cells were cultured on 48-well plates for 6 days in proinflammatory conditions (TNF-α/IL-1a/C1q), anti-inflammatory conditions (IL-4, IL-10, or TGF-β1), and in non-stimulatory conditions (culture medium). Data shown as mean chemokine concentration ± SD. Normality of the distribution was checked with Shapiro–Wilk test. For comparisons between groups, Mann–Whitney U test was used and differences were considered significant for *p* values < 0.05.

**Figure 4 biomedicines-10-01769-f004:**
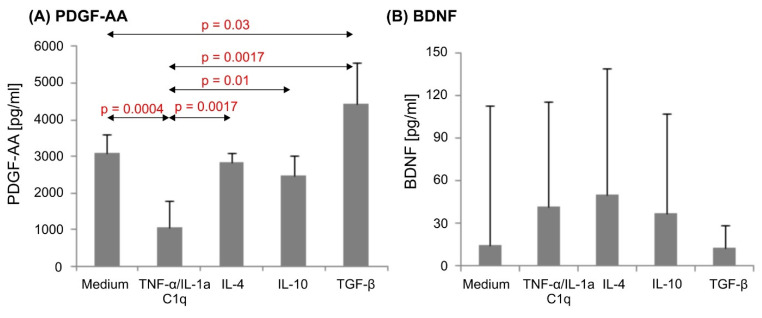
PDGF-AA (**A**) and BDNF (**B**) production in human astrocyte cultures. Results from 8 experiments, conducted on primary astrocyte cells collected from 4 human donors. Cells were cultured on 48-well plates for 6 days in proinflammatory conditions (TNF-α/IL-1a/C1q), anti-inflammatory conditions (IL-4, IL-10, or TGF-β1), and in non-stimulatory conditions (culture medium). Data shown as mean PDGF-AA (**A**) or BDNF (**B**) concentration ± SD. For PDGF-AA results, statistical analysis was performed with parametric one-way ANOVA test followed by post-hoc Tukey’s test. Analysis of BDNF results was carried out with non-parametric Kruskal–Wallis test. Normal distribution within groups was checked with Shapiro–Wilk test, *p* values < 0.05 were considered as significant.

## Data Availability

Not applicable.

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
