# Peer review of "Human Primary Astrocytes Differently Respond to Pro- and Anti-Inflammatory Stimuli"

_biomedicines, 2022, doi:10.3390/biomedicines10081769_

Round 1
Reviewer 1 Report
The present study aims to evaluate the production of inflammatory chemokines and neurotrophic factors in human astrocytes by different types of stimulation. The authors described the aims and goals very clearly, followed by an explanatory introduction and discussion. However, the study could have a bigger impact if other methods would be applied, like proteomics to determine the protein expression, immunohistochemistry to localize their expression and also a co-culture model to mimic a more realistic microenvironment.
1- In figure 1: The authors present the results of GFAP expression in astrocytes, only figure 1C is mentioned in the text. Consider a better figure description and also more details/explanation of the figure including in the results section. The flow cytometry plot for population 1: it is not clear the population, the debri/dead cells are very mixed with the selected population. Did the authors have a plot where the cell population is more easily distinguished from the debris? I would also recommend adding a control cell type that is known to not express the GFAP marker (https://doi.org/10.1161/ATVBAHA.120.314078).
2- The authors showed the upregulation of CD83 and HLA-DR, however only one donor is present in the figure. I would recommend having a more detailed figure including all the donors or consider a supplementary figure. For the marker expression, it would be interesting to have immunohistochemistry showing the intensity staining for the referred markers.
3- The authors should have a better organization of the figures, some of the figures are labeled with capital letters, others with small case letters. The authors should consider adding a more detailed explanation about the figure and also about the statistics used.
4- On figure 3 what was the statistical method that the authors used? I would recommend applying a different statistical method and expressing the error in SD instead of SEM.
5- How were the cytokines selected? The authors should consider explaining their selection during the discussion section.
6- In the discussion section, when the authors discuss their results, they should include a reference to the figures.
7- On the discussion section, starting on line 264, the authors are sharing results from which there is no reference on the manuscript, therefore, a proper reference is missing for these results.
8- Did the authors consider having a 3D model of the blood brain barrier? The endothelial cells and pericytes also plays an important role during neuroinflammation, expressing a different secretome (PMID: 29751771), therefore, how the astrocytes react the external stimulus from their microenvironment can be different then when adding directly to the cells.
9- The abbreviations during the manuscript should be better organized
10- On the material and methods section 2.1 it is not clear what the 418-22-week-old donor means. I would suggest rephrasing or give more information.
11- I believe there are some inconsistencies in terms of results description, like in line 176, where the authors mentioned “Figure 3b” I believe it is referring to Figure 4, that should be a b, but is a labeled as a “c”.
Author Response
We would like to thank the reviewer fo his suggestions. Please find the specific answers to the reviewer’s questions below.
Reviewer :
The present study aims to evaluate the production of inflammatory chemokines and neurotrophic factors in human astrocytes by different types of stimulation. The authors described the aims and goals very clearly, followed by an explanatory introduction and discussion. However, the study could have a bigger impact if other methods would be applied, like proteomics to determine the protein expression, immunohistochemistry to localize their expression and also a co-culture model to mimic a more realistic microenvironment.
Answer:
Of course more global approaches utilizing whole proteome analysis would provide more complex data. In our research, we focused on the analysis of the parameters important for astrocytes’ interactions with immune system cells. We measured the expression of GFAP protein as an astrocytes’ marker and we observed its increase in pro-inflammatory cytokines’ milieu what suggest GFAP synthesis and increased cell proliferation activity. We have analyzed the expression of selected co-stimulatory molecules on astrocytes to assess their potential in antigen presentation and T-cells’ activation as we believe, this cross-talk is important in autoimmune diseases like Multiple Sclerosis. Analysis o IL-1β as well as chemokines’ secretion was performed to characterize the proinflammatory properties of astrocytes after their exposition to cytokine stimuli. Production of IL-1β by these cells was suggested to be the feature of proinflammatory and neurotoxic astrocytes. We choose various chemokines as a proinflammatory proteins implicated in regulation of immune cells’ migration to visualize astrocytes’ potential in recruiting immune cells to the inflamed area in the brain, especially that astrocytes are one of the cellular components of blood-brain barrier. (Curr Opin Neurobiol. 2022 May 8;74:102550. doi: 10.1016/j.conb.2022.102550)
Reviewer:
1- In figure 1: The authors present the results of GFAP expression in astrocytes, only figure 1C is mentioned in the text. Consider a better figure description and also more details/explanation of the figure including in the results section. The flow cytometry plot for population 1: it is not clear the population, the debri/dead cells are very mixed with the selected population. Did the authors have a plot where the cell population is more easily distinguished from the debris? I would also recommend adding a control cell type that is known to not express the GFAP marker (https://doi.org/10.1161/ATVBAHA.120.314078).
Answer:
We have mentioned all figures 1A-B in the text. We have combined previous Fig 1A with B. Additionally, we have modified the description of the figure to „A) Expression of glial fibrillary acidic protein (GFAP)—2D graphs of GFAP-positive astrocyte cells and gating strategy. B) Cytokine-induced changes in GFAP expression—representative histograms from 1 donor. Red-filled histograms represent cells stained intracellulary with isotype control antibodies, blue histograms represents cells stained with GFAP-specific antibodies conjugated with PE. Combined results from 4 donors showing increased GFAP level are presented on the graph as changes in MFI (median fluorescence intensity)”.
We have also modified the result section corresponding to this figure: „To confirm that the analyzed cells were astrocytes, we measured the intracellular level of the astrocyte marker—the GFAP protein—by flow cytometry. All analyzed cells showed a high intracellular GFAP protein level (Figure 1A). GFAP increased in astrocytes exposed to the microglia proinflammatory cytokine mixture (TNF-α/IL-1a/C1q) compared to unstimulated cells (p = 0.02) (Figure 1B) [16]. We did not notice any significant changes in GFAP level in cells stimulated with other cytokines. Cells cultured in medium alone were used as a control”.
We have changed the plots to more clear ones.
Thank you for the suggestion according to the use of other control cell type, however now this would be related to the preparation of all experiments ones more from the beggining, what would be very time and money consuming.
Reviewer:
2- The authors showed the upregulation of CD83 and HLA-DR, however only one donor is present in the figure. I would recommend having a more detailed figure including all the donors or consider a supplementary figure. For the marker expression, it would be interesting to have immunohistochemistry showing the intensity staining for the referred markers.
Answer:
Results for other donors are included as a supplementary figure, according to the reviewer’s suggestion.
We agree with the reviewer that it would be interesting to have immunohistochemistry showing the intensity staining for the referred markers, however we were focused on the quantitative rather that gualitative analysis. We are considering adding such experiments in the future.
Reviewer:
3- The authors should have a better organization of the figures, some of the figures are labeled with capital letters, others with small case letters. The authors should consider adding a more detailed explanation about the figure and also about the statistics used.
Answer:
All inconsistancies in figures were corrected. We have also added more detailed description of the figures and statistics used.
Reviewer:
4- On figure 3 what was the statistical method that the authors used? I would recommend applying a different statistical method and expressing the error in SD instead of SEM.
Answer:
We have used Shapiro-Wilk test to check the normality of the distribution. For comparisons between groups Mann Whitney U test was used.
The error expression was changed for SD instead of SEM according to reviewer’s suggestion.
Reviewer:
5- How were the cytokines selected? The authors should consider explaining their selection during the discussion section.
Answer:
The cytokines were selected based on the literature related to astrocytes’ phenotype. We have added more information according to our cytokines’ selection in the Material and Method section: „Astrocytes were stimulated in culture medium with a cocktail of proinflammatory cytokines: rhTNF-α (30 ng/ml, R&D Systems), rhIL-1a (3 ng/ml, R&D Systems), hC1q (400 ng/ml, MyBiosource). These cytokines are released by microglia cells and were reported to have potential in induction of proinflammatory and neurotoxic astrocytes’ phenotype [16]. For alternative stimulation, cells were cultured in medium containing rhIL-10 (10 ng/ml, R&D Systems), IL-4 (1ng/ml R&D Systems) or TGF-β (10 ng/ml, R&D Systems) as these cytokines possess anti-inflammatory potential and were reported to be present in the brain during inflammation [17,18].”
Reviewer :
6- In the discussion section, when the authors discuss their results, they should include a reference to the figures.
Answer:
We have included references to the our figures in the discussion section, according to the reviewer’s suggestion.
Reviewer :
7- On the discussion section, starting on line 264, the authors are sharing results from which there is no reference on the manuscript, therefore, a proper reference is missing for these results.
Answer:
We have added missed reference.
Reviewer :
8- Did the authors consider having a 3D model of the blood brain barrier? The endothelial cells and pericytes also plays an important role during neuroinflammation, expressing a different secretome (PMID: 29751771), therefore, how the astrocytes react the external stimulus from their microenvironment can be different then when adding directly to the cells.
Answer:
We agree with the reviewer’s statement. The concept is worth to be explored in future experiments. However, our goal is to check the potential of cytokines-treated astroocytes to shape the activity of immune response, which is an important process in various neuroinflammatory disorders. Morover, there are serious limitations in obtaining primary human endothelial cells, as the viability of these cells is very low, and often they have a high rate of death during transport.
Reviewer :
9- The abbreviations during the manuscript should be better organized
Answer:
The organization of the abbreviations has been improved.
Reviewer :
10- On the material and methods section 2.1 it is not clear what the 418-22-week-old donor means. I would suggest rephrasing or give more information.
Answer:
The sentence has been modified to „Cerebral cortex astrocytes isolated from the human brain were obtained from four 18 - 22-week-old donors…”.
Reviewer :
11- I believe there are some inconsistencies in terms of results description, like in line 176, where the authors mentioned “Figure 3b” I believe it is referring to Figure 4, that should be a b, but is a labeled as a “c”.
The mistake has been corrected.
Answer:
Reviewer 2 Report
This article compares how astrocytes differentially respond to proinflammatory and anti-inflammatory stimuli.
My general impression of this article is that it is well-written, and addresses an important topic of astroglial response to different types of stimuli. Results indicate that astrocytes express different chemokines and neurotrophic factors after proinflammatory stimulation compared with anti-inflammatory cytokine stimulation.
Suggested edits below should be considered minor revisions.
1. There are many places in the abstract and manuscript where spacing is missing between words. For example, in the first sentence of the abstract, ‘shownthat’ should be corrected to ‘shown that’ and in the last sentence of the abstract ‘toinflammatory’ should be correct to ‘to inflammatory.’ Please correct the many additional spacing issues throughout the manuscript.
2 2. Introduction. On line 44, please add a reference for the sentence that begins, ‘It is suggested that they may also act as antigen presenting cells….the CNS.’
3 3. Introduction. On page 2, line 56-58, in the sentence that starts, ‘the mechanisms translating injury-derived’ and ends with ‘remain largely uncovered.’ I would recommend perhaps changing ‘uncovered’ to ‘unknown.’
4 4. Introduction. On page 2, line 64, ‘thesecretory’ should be corrected to ‘the secretory and not underlined.
5 5. Material and Methods. In Section 2.1, in the first sentence, please remove the first hyphen from ‘418-22-week old donors.’ Also please correct ‘donorsand’ to ‘donors and.’
66. Discussion. On Page 7, line 189, please add a reference to the end of the sentence.
77. Discussion. On Page 7, line 202, please add a reference to the end of the sentence.
88. Discussion. On page 7, line 207, please spell out ‘EAE’ on first usage.
99. Conclusions. On page 10, line 364, please remove the second period at the end of the sentence.
Author Response
We would like to thank the reviewer for his suggestions. Please find the specific answers below.:
Reviewer:
1 There are many places in the abstract and manuscript where spacing is missing between words. For example, in the first sentence of the abstract, ‘shownthat’ should be corrected to ‘shown that’ and in the last sentence of the abstract ‘toinflammatory’ should be correct to ‘to inflammatory.’ Please correct the many additional spacing issues throughout the manuscript.
Answer:
All the missed spacing have been corrected along the whole mauscript.
Reviewer:
2. Introduction. On line 44, please add a reference for the sentence that begins, ‘It is suggested that they may also act as antigen presenting cells….the CNS.’
Answer:
The reference has been added.
Reviewer:
3 3. Introduction. On page 2, line 56-58, in the sentence that starts, ‘the mechanisms translating injury-derived’ and ends with ‘remain largely uncovered.’ I would recommend perhaps changing ‘uncovered’ to ‘unknown.’
Answer:
The sentence has been modified according to the reviewer’s suggestion.
Reviewer:
4 4. Introduction. On page 2, line 64, ‘thesecretory’ should be corrected to ‘the secretory and not underlined.
Answer:
The part of the sentence has been corrected.
Reviewer:
5 5. Material and Methods. In Section 2.1, in the first sentence, please remove the first hyphen from ‘418-22-week old donors.’ Also please correct ‘donorsand’ to ‘donors and.’
Answer:
The first sentence from the Material and Methods section has been corrected.
Reviewer:
- Discussion. On Page 7, line 189, please add a reference to the end of the sentence.
Answer:
The reference has been added.
Reviewer:
- Discussion. On Page 7, line 202, please add a reference to the end of the sentence.
Answer:
The reference has been added.
Reviewer:
- Discussion. On page 7, line 207, please spell out ‘EAE’ on first usage.
Answer:
The abbreviation EAE has been explained with the first usage, according to the reviewer’s suggestion.
Reviewer:
- Conclusions. On page 10, line 364, please remove the second period at the end of the sentence.
Answer:
The second period has been removed.
Round 2
Reviewer 1 Report
The authors addressed the majority of my questions. However, I believe this study can be improved significantly.
1- Primary cultured cells have a higher percentage of other cell contaminants. Therefore, adding an extra marker, for example an endothelial cell marker CD31 or CD144, to determine the purity of the culture would benefit the study.
2- To prove the functional phenotype of the human astrocytes in response to stimulatory factors, the authors should add immunohistochemistry from selected markers to determine the cell phenotype in response to the cytokines.
3- Aquaporin-4 expression changes during inflammation (Galea, I. The blood–brain barrier in systemic infection and inflammation. Cell Mol Immunol 18, 2489–2501 (2021). https://doi.org/10.1038/s41423-021-00757-x) and it is known to be a specific marker for astrocytes (Hubbard JA, Szu JI, Binder DK. The role of aquaporin-4 in synaptic plasticity, memory and disease. Brain Res Bull. 2018 Jan;136:118-129. doi: 10.1016/j.brainresbull.2017.02.011. Epub 2017 Mar 6. PMID: 28274814.), therefore the authors should include in the study how the expression changes before and after the cytokine stimulation.
4- Astrocytes control the BBB integrity. Endothelial cells (EC) are one of the principal elements that constitutes the blood brain barrier. During inflammation the EC changes their signalling pathways, their cellular traffic and also an increase in permeability including immune cells. (Galea, I. The blood–brain barrier in systemic infection and inflammation. Cell Mol Immunol 18, 2489–2501 (2021). https://doi.org/10.1038/s41423-021-00757-x; Daneman R, Prat A. The blood-brain barrier. Cold Spring Harb Perspect Biol. 2015;7(1):a020412. Published 2015 Jan 5. doi:10.1101/cshperspect.a020412). Therefore, the authors should include a 2D culture with endothelial cells in order to observe if the astrocytes express the same phenotype as described in the present study, since the immune response also depends on the endothelial cells reacting to the stimulus.
5- On figure 3 and 4, the error bars are extremely high. How many parallels are present in the study? The authors should consider adding more parallels to increase the statistics confidence and also significance.
6- Figures can be improved, be more uniform and have the same color, font type and size.
Author Response
Below are our detailed comments to the new suggestions:
- Primary cultured cells have a higher percentage of other cell contaminants. Therefore, adding an extra marker, for example an endothelial cell marker CD31 or CD144, to determine the purity of the culture would benefit the study.
Why the Reviewer suggested endothelial markers, not an oligodendrocyte or microglia markers? We suppose this cells might be even more frequent as contaminants than endothelial cells.
- To prove the functional phenotype of the human astrocytes in response to stimulatory factors, the authors should add immunohistochemistry from selected markers to determine the cell phenotype in response to the cytokines.
There are several methods to prove changes in functional phenotype of astrocytes. Their secretory activity can be modified, and also expression of various surface receptors. We believe that immunohistochemistry will add additional valuable information, however the expression of intracellular and surface molecules by astrocytes is also highly informative.
- Aquaporin-4 expression changes during inflammation (Galea, I. The blood–brain barrier in systemic infection and inflammation.Cell Mol Immunol 18, 2489–2501 (2021). https://doi.org/10.1038/s41423-021-00757-x) and it is known to be a specific marker for astrocytes (Hubbard JA, Szu JI, Binder DK. The role of aquaporin-4 in synaptic plasticity, memory and disease. Brain Res Bull. 2018 Jan;136:118-129. doi: 10.1016/j.brainresbull.2017.02.011. Epub 2017 Mar 6. PMID: 28274814.), therefore the authors should include in the study how the expression changes before and after the cytokine stimulation.
We have read all the papers mentioned by the reviewer, they were all review articles about blood-brain barier composition and functions, not about AQP-4 in inflammatory astrocytes, as he tried to convienced us. First of the mentioned work in fact stated that AQP-4 increases in inflammation and refers to the research made on rats, where the increase of AQP-4 mRNA was reported in brain after LPS injections. However, in our opinion it is not clear if this finding is related to increase in AQP-4 expression on astrocytes but rather to astrocyte proliferation.
- Astrocytes control the BBB integrity. Endothelial cells (EC) are one of the principal elements that constitutes the blood brain barrier. During inflammation the EC changes their signalling pathways, their cellular traffic and also an increase in permeability including immune cells. (Galea, I. The blood–brain barrier in systemic infection and inflammation.Cell Mol Immunol 18, 2489–2501 (2021). https://doi.org/10.1038/s41423-021-00757-x; Daneman R, Prat A. The blood-brain barrier. Cold Spring Harb Perspect Biol. 2015;7(1):a020412. Published 2015 Jan 5. doi:10.1101/cshperspect.a020412). Therefore, the authors should include a 2D culture with endothelial cells in order to observe if the astrocytes express the same phenotype as described in the present study, since the immune response also depends on the endothelial cells reacting to the stimulus.
We have answered to the similar suggestion previously: „We agree with the Reviewer’s statement. The concept is worth to be explored in future experiments. However, our goal is to check the potential of cytokines-treated astrocytes to shape the activity of immune response, which is an important process in various neuroinflammatory disorders. Morover, there are serious limitations in obtaining primary human endothelial cells, as the viability of these cells is very low, and often they have a high rate of death during transport.”
- On figure 3 and 4, the error bars are extremely high. How many parallels are present in the study? The authors should consider adding more parallels to increase the statistics confidence and also significance.
As stated in figures description results comes from cells from 4 donors cultured in 2 parallels. This were the primary cells’ cultures and the individual diversity may occure. However, the statisticall analysis in spite of such diversity has shown significance and the exact values of p are given on graphs.
- Figures can be improved, be more uniform and have the same color, font type and size.
We can improve figures according to the suggestion.
The second reviewer gave as minor revision during the first round of revision, and have no comments and suggestions after our answers and improvements
In our opinion the reviewer has the chance to write down all his statements in the first round of revision, and in the second he shouldn’t give additional new suggestions only to the new corrections.